# One year cross-sectional study in adult and neonatal intensive care units reveals the bacterial and antimicrobial resistance genes profiles in patients and hospital surfaces

**Ana Paula Christoff**[1], **Aline Fernanda Rodrigues Sereia**[1], **Giuliano Netto Flores Cruz**[1], **Daniela Carolina de Bastiani**[1], **Vanessa Leitner Silva**[1], **Camila Hernandes**[2], **Ana Paula Metran Nascente**[2], **Ana Andrea dos Reis**[2], **Renata Gonçalves Viessi**[2], **Andrea dos Santos Pereira Marques**[2], **Bianca Silva Braga**[2], **Telma Priscila Lovizio Raduan**[2], **Marines Dalla Valle Martino**[2], **Fernando Gatti de Menezes**[2], **Luiz Felipe Valter de Oliveira**[1]*

**1** BiomeHub, Florianopolis, Brazil, **2** Albert Einstein Israelite Hospital, São Paulo, Brazil

* felipe@biomehub.com

## Abstract

Several studies have shown the ubiquitous presence of bacteria in hospital surfaces, staff, and patients. Frequently, these bacteria are related to HAI (healthcare-associated infections) and carry antimicrobial resistance (AMR). These HAI-related bacteria contribute to a major public health issue by increasing patient morbidity and mortality during or after hospital stay. Bacterial high-throughput amplicon gene sequencing along with identification of AMR genes, as well as whole genome sequencing (WGS), are biotechnological tools that allow multiple-sample screening for a diversity of bacteria. In this paper, we used these methods to perform a one-year cross sectional profiling of bacteria and AMR genes in adult and neonatal intensive care units (ICU and NICU) in a Brazilian public, tertiary hospital. Our results showed high abundances of HAI-related bacteria such as *S. epidermidis*, *S. aureus*, *K. pneumoniae*, *A. baumannii* complex, *E. coli*, *E. faecalis*, and *P. aeruginosa* in patients and hospital surfaces. Most abundant AMR genes detected throughout ICU and NICU were *mecA*, *bla*$_{CTX-M-1\ group}$, *bla*$_{SHV-like}$, and *bla*$_{KPC-like}$. We found that NICU environment and patients were more widely contaminated with pathogenic bacteria than ICU. Patient samples, despite the higher bacterial load, have lower bacterial diversity than environmental samples in both units. Finally, we also identified contamination hotspots in the hospital environment showing constant frequencies of bacterial and AMR contamination throughout the year. Whole genome sequencing (WGS), 16S rRNA oligotypes, and AMR identification allowed a high-resolution characterization of the hospital microbiome profile.

## Introduction

Hospital indoor environment including surfaces, staff and patients are the focus of major recent investigations regarding the microbiome and its bacterial composition [1–4]. This

**Data Availability Statement:** All sequence data are deposited in NCBI BioProject PRJNA604445

**Funding:** The funders BiomeHub and Albert Einstein Israelite Hospital (HIAE) provided support in form of salaries for the authors and were responsible for the project experimental design approval. HIAE have an additional role in defining sample collection points, but the analysis, the decision to publish and the manuscript preparation was independently performed by the researchers. The specific role of each author are included in the 'author contributions' section.

**Competing interests:** I have read the journal's policy and the authors of this manuscript have the following competing interests: APC, AFRS, GNFC, DCB, VLS, and LFVO are currently full-time employees of BiomeHub (SC, Brazil), a research and consulting company specialized in microbiome technologies. This does not alter our adherence to PLOS ONE policies on sharing data and materials.

extensive investigations became possible due to the dissemination of culture-independent microbiology methods using next-generation DNA sequencing directly from the collected samples. Culture-independent methods like DNA sequencing, are less time-consuming and less skewed to detect only specific cultivable microorganisms compared to traditional microbiological methods, allowing large-scale screening of microorganisms, including those that will not grow well in conventional microbiology conditions [5–9].

Several studies already demonstrated the presence and persistence of bacterial pathogens in hospital surfaces, like *Enterococcus*, *Staphylococcus aureus*, *Acinetobacter*, *Escherichia coli*, *Klebsiella*, *Pseudomonas aeruginosa* and *Serratia marcescens* [10–12]. These microorganisms, among others listed by CDC–Centers for Disease Control and Prevention (https://www.cdc.gov/hai/index.html)—are known as nosocomial pathogens related to Healthcare-associated infections (HAI). These pathogenic bacteria can also harbor multidrug-resistance genes (MDR) and confer a broad spectrum of antimicrobial resistance (AMR)—a matter of major concern in public health, leading to increased patient morbidity and mortality during or after hospital stay [13,14].

Healthcare-associated infections are critical issues for adult and neonatal intensive care units (ICU and NICU, respectively), increasing the patient hospital stay, mortality rates and medical costs [14]. Especially in developing countries like Brazil, patients were dramatically affected by nosocomial infections and, despite the few studies available, similar rates for ICU and NICU infections were reported, but with presumed case fatality from 12 to 50% in neonates [13,15,16]. Different hospitals have different bacterial profiles in their intensive care units, the only common factors seem to be that patients in intensive care units were more prone to bacterial infections given their compromised health state and that ICUs and NICUs have high bacterial contamination rates in inanimate surfaces and equipment [4,17–19].

Hospital contaminated environments, including surfaces and staff, are well recognized reservoirs and transmission sources for HAI-related pathogens [2,12,17,20–25]. In this scenario, the study of hospital microbiome has substantial implications in the healthcare system: the continuous monitoring of hospital environment, bacterial tracking, and detailed epidemiological investigations may contribute to decrease HAI rates and improve healthcare system quality. In this study, by monthly collecting DNA samples from environment and patients, we investigated the microbiome of adult and neonatal intensive care units from a local public hospital over a one-year period. We perform exhaustive molecular assessment to understand the profiles of bacterial abundances as well as antimicrobial resistance genes from the healthcare environment and patient samples.

## Material and methods

### Study design

This study was performed in a tertiary public hospital in the city of São Paulo, Brazil. A twelve-month project was designed comprising adult intensive care unit (ICU) and neonatal intensive care unit (NICU) samples from environmental surfaces and patients (Table 1). From August-2018 to July-2019, patient beds and common use areas were sampled each month, as well as

**Table 1. Study samples along one year.**

|  | ICU SAMPLES | NICU SAMPLES |
|---|---|---|
| **PATIENT** | 138 | 111 |
| **BEDS** | 676 | 598 |
| **COMMON ENVIRONMENT** | 248 | 207 |

patient nasal and rectal swabs. For neonatal patients, fresh stool swabs were collected instead of rectal (as an ethics committee requirement for the project). Bed environmental surface samples were collected from medical and hospital equipment, furniture, critical structure points, and bed accessories–all characterized as high contact surfaces. Detailed collection sites are described in S1 Table. All patient samples collected were from the same beds included in the environmental sampling. Common areas include nurse stations and common use medical equipment. Additionally, some bacterial isolates were obtained by the hospital microbiology laboratory for further genomic analysis. This project was approved by the Albert Einstein Israelite Hospital ethics committee (number 2.585.209). All participants were informed about the study aims and sampling was carried out upon a signature of an informed consent by the patient or a legal representative.

## Sample collection and DNA extraction

Samples were collected using a sterile cotton swab, without transport media. The swab was moistened with a sterile saline solution (0.9% NaCl) prior to sample collection. The swabs were transported to the laboratory facilities at room temperature and processed within a maximum of 24h after sample collection. Bacterial DNA from the samples was obtained through a thermal lysis (96˚C– 10 min) followed by a purification step with AMPure XP Magnetic Beads (Beckman Coulter, USA). Negative controls were included in each lysis and DNA extraction batch.

## Library preparation and DNA sequencing

We performed amplicon sequencing library preparation for bacteria using the V3/V4 16S rRNA gene primers 341F and 806R [26,27] in a two-step equivolumetric PCR protocol [28]. The first PCR was performed with V3/V4 universal primers containing a partial Illumina adaptor, based on TruSeq structure (Illumina, USA) that allows a second PCR with the indexing sequences similar to procedures described previously [29]. Here, we add unique dual-indexes per sample in the second PCR. Two microliters of individual sample DNA were used as input in each first PCR reaction. The PCR reactions were carried out using Platinum Taq (Invitrogen, USA) with the conditions: 95˚C for 5 min, 25 cycles of 95˚C for 45s, 55˚C for 30s and 72˚C for 45s and a final extension of 72˚C for 2 min for PCR 1. In PCR 2 the conditions were 95˚C for 5 min, 10 cycles of 95˚C for 45s, 66˚C for 30s and 72˚C for 45s and a final extension of 72˚C for 2 min. All PCR reactions were performed in triplicate. The final PCR reactions were cleaned up using AMPureXP beads (Beckman Coulter, USA) and an equivalent volume of each sample was added in the sequencing pool. At each batch of PCR, a negative reaction control was included (CNR). The final DNA concentration of the libraries pool was estimated with Picogreen dsDNA assays (Invitrogen, USA) and then diluted for accurate qPCR quantification using KAPA Library Quantification Kit for Illumina platforms (KAPA Biosystems, USA). The sequencing pool was adjusted to a final concentration of 11 pM (for V2 kits) or 18 pM (for V3 kits) and sequenced in a MiSeq system (Illumina, USA), using the standard Illumina primers provided by the manufacturer kit. Single-end 300 cycle runs were performed using V2x300, V2x500 or V3x600 sequencing kits (Illumina, USA) with sample coverages set to 45,000 reads per sample in all sequencing runs (S2 Table).

## Sequencing data analysis

The sequenced reads obtained were processed using an *in-house* developed bioinformatics pipeline described below (BiomeHub, Brazil–hospital_miccrobiome_rrna16s:v0). Illumina FASTQ files had the primers trimmed and their accumulated error evaluated [28]. Only one

mismatch is allowed in the primer sequence that should be present at the beginning of the read. The whole read sequence is discarded if this criterion is not met. Reads were analyzed with the Deblur package [30] to remove possible erroneous reads and then identical read sequences were grouped into oligotypes (clusters with 100% identity). The sequence clustering with 100% identity provides a higher resolution for the amplicon sequencing variants (ASVs), also called sub-OTUs (sOTUs) [31]—herein denoted as oligotypes. Next, VSEARCH [32] was used to remove chimeric amplicons.

An additional filter was implemented to remove oligotypes below the frequency cutoff of 0.2% in the final sample counts. We also implemented a negative control filter, since hospital microbiome generally are low biomass samples [28]. In each processing batch, we used negative controls for the DNA extraction and PCR reactions. If any oligotypes are recovered in the negative control results, they are checked against the samples and automatically removed from the results only if their abundance (in number of reads) are no greater than two times their respective counts in the sample. The remaining oligotypes in the samples are used for taxonomic assignment with the BLAST tool [33] against a reference genomic database (encoder-ef16s_rev6_190325). This *in-house* database was constructed with complete and draft bacterial genomes, focused on clinically relevant bacteria, obtained from NCBI. It is composed of 11,750 sequences including 1,843 different bacterial taxonomies.

Taxonomy was assigned to each oligotype using a lowest common ancestor (LCA) algorithm. If more than one reference can be assigned to the same oligotype with equivalent similarity and coverage metrics (e.g. two distinct reference species mapped to oligotype "A" with 100% identity and 100% coverage), the taxonomical assignment algorithm leads the taxonomy to the lowest level of possible unambiguous resolution (genus, family, order, class, phylum or kingdom), according to the similarity thresholds previously established [34].

After quality check of sequencing yield (S2 Table), the resulting oligotype tables, analogous to traditional OTU tables, were processed and normalized as previously described [28]. Oligotype sequences served as input for FastTree 2.1 software [35] to construct phylogenetic trees and allow beta-diversity analysis with UniFrac distances [36]. Additional analyses were performed using R (version 3.6.0) and the Phyloseq package [37]. When suited, non-parametric comparisons were performed using Kruskall-Wallis and Wilcoxon tests implemented in R [38]. Alpha diversity analysis was performed using the "plot_richness" function in Phyloseq.

## RGene–antimicrobial resistance gene analysis

A panel comprising relevant β-lactamases, Vancomycin, Methicillin and Colistin antimicrobial resistance genes in Brazilian scenario was tested: $bla_{CTX-M-1}$ group, $bla_{CTX-M-2}$ group, $bla_{CTX-M-8}$ group, $bla_{CTX-M-9}$ group, $bla_{GES-like}$, $bla_{IMP-like}$, $bla_{KPC-like}$, $bla_{NDM-like}$, $bla_{SHV-like}$, $bla_{SPM-like}$, $bla_{VIM-like}$, $bla_{OXA-143-like}$, $bla_{OXA-23-like}$, $bla_{OXA-48-like}$, $bla_{OXA-51-like}$, $bla_{OXA-72-like}$, *vanA*, *vanB*, *mecA* and *MCR1*. The detection was performed using Real-Time PCR with QSY hydrolysis probes labeled with FAM®, VIC® and NED® (Applied Biosystems, USA). To test primer and probe efficiency we used bacterial strains containing the resistance genes of interest (kindly provided by Prof. Dr. Ana Cristina Gales). These bacterial strains were also included in each PCR run as positive controls.

Real Time PCR reactions were carried out using 10 μL of final volume per sample, containing 2 μL of the same previously sequenced DNA samples, 0.2 U Platinum Taq, 1 X Buffer, 3 mM MgCl2, 0.1 mM dNTP, 0.12 X ROX^TM and 0.2 μM of each forward and reverse specific primer following the thermal conditions: 95˚C for 5 min with 35 cycles of 95˚C for 15s, 60˚C for 30s and 72˚C for 30s. Negative and positive reaction controls were included in all the assays. All the samples were analyzed in experimental triplicate. Real Time reactions were

performed in an ABI 7500 Fast Real-Time PCR System (Applied Biosystems, USA). Samples were considered positive when at least two of the experimental replicates were below the quantification cycle 33 using an experimental threshold of 0.05.

### Bacterial genomes

Bacterial isolates from the hospital routine surveillance swabs, during the same week that the molecular biology collection took place, were selected for whole genome sequencing. Isolates were processed as described above for DNA extraction, and the sequencing library preparation was performed using Nextera Flex with CD indexes (Illumina, USA), according to manufacturer instructions. Samples were sequenced in a MiSeq system (Illumina, USA), in paired-end 150 pb configuration, with coverage of 1 million reads per sample.

Sequenced genomes were analyzed using A5 [39], SPAdes [40] and Prokka [41]. Species identification and average nucleotide identity (ANI) were performed using the JSpecies platform [42]. The whole genome sequence was used to identify the Multi Locus Sequence Typing (MLST 2.0) as previously described [43]. Clonality analysis was performed among bacteria of same species using NDTree [44] to assess genome single nucleotide variants (SNVs) and evaluating ANI values using JSpecies ANIm calculations [42].

## Results

### Samples and high-throughput amplicon sequencing

A total of 1,978 samples were collected from August-2018 to July-2019 (Tables 1 and S1). Twelve sequencing runs were performed resulting in 78,840,894 reads, with an average and standard deviation of 6,570,075 ± 813,906 reads per run and 30,942 ± 3,468 reads by sample (S2 Table). In the $\log_{10}$ scale, patient and healthcare environment median reads in ICU and NICU reflects the scales of microbial load in the samples (Fig 1). Some months showed lower medians, suggesting samples with lower bacterial load, however, the annual profile is pretty similar.

Considering quality reads that passed through our bioinformatics pipeline, 98.74% of reads could be classified as bacteria (kingdom), 97.84% were classified at family level, 89.14% at genus level, and 67.28% at species level. Of the global sequencing results, 40.96% of the reads could be identified in a HAI-related Bacteria group (here denoted HAIrB group: *Acinetobacter baumanii* complex, *Escherichia coli*, *Enterococcus faecalis*, *Enterococcus faecium*, *Klebsiella pneumoniae*, *Proteus mirabilis*, *Pseudomonas aeruginosa*, *Staphylococcus aureus* and *Staphylococcus epidermidis*), corresponding to 60.87% of the results at species level.

Throughout the assessed year, 19.79% of the identified bacterial DNA belonged to the genus *Staphylococcus* (11.79% *S. epidermidis*, 3.03% *S. aureus*, 4.97% other species). Other HAIrB had the following frequencies: 7.39%—*A. baumannii* complex, 7.06%—*E. coli*, 3.27%—*Enterococcus* and 5.73%—*K. pneumoniae*. Thus, we can state that the genus *Staphylococcus* and the species *S. epidermidis* have predominant observed proportions in this hospital microbiome throughout this year of analysis.

### Bacterial diversity analysis

Alpha diversity analysis for both ICU and NICU showed higher and more variable values of Shannon index in the healthcare environment than in patient samples along the year (Fig 2A). Similar pattern was observed when looking at specific sampling sites within units (Fig 2B). Longitudinal profiles of alpha diversity along the year in different hospital locations and

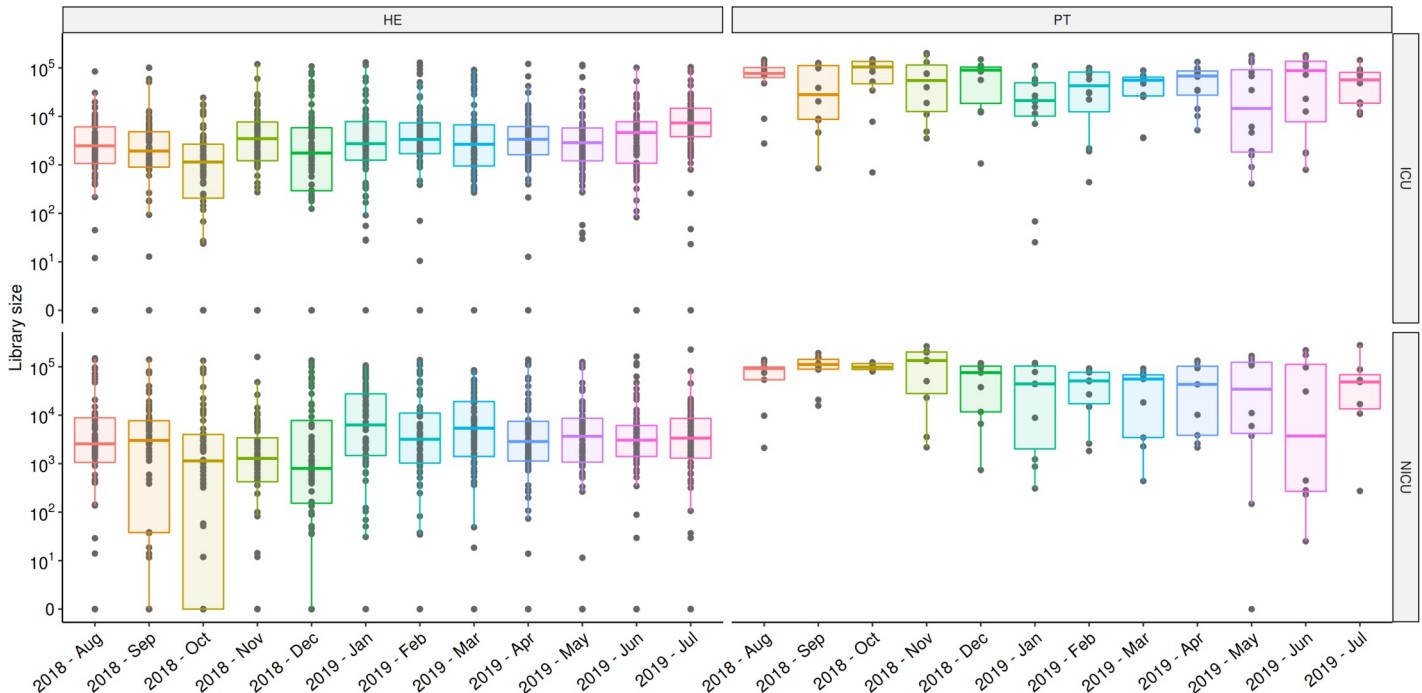

**Fig 1. Total sequenced reads in ICU and NICU along the year.** Total sequenced reads (library size) were represented by boxplots with the median bacterial distribution for each month of samples collected in ICU (upper panel) and NICU (lower panel). Healthcare environment (HE–left panel) and patients (PT–right panel) were depicted separately showing their particular variabilities along the year. Boxplots represents median distributions of the total reads obtained in samples, considering $\log_{10}$ scale.

specific collection sites are shown in S1 Fig, revealing highly variable values even among equivalent sample sites within and between months.

Weighted UniFrac analysis showed similar distribution patterns among samples across the year (Fig 2C), without clear separation between months, but with a slightly differential distribution of samples from August to November 2018, December 2018 to February 2019, and April to July 2019 (S2 Fig). ICU and NICU samples have similar beta-diversity profiles (Fig 2D), and only patient samples seem to cluster more closely (Fig 2E).

## Bacterial profiling in the hospital

Several bacterial taxonomies were identified in this study and classified into phylum, family, genus, and species levels. Proteobacteria are by far the most predominant bacterial phylum found in the results of both ICU and NICU, followed by Firmicutes, Actinobacteria, and Bacteroidetes. The most abundant families found were: Enterobacteriaceae, Staphylococcaceae, Moraxellaceae, Pseudomonadaceae, Corynebacteriaceae, Streptococcaceae, Enterococcaceae, Peptoniphilaceae, Prevotellaceae and Bacteroidaceae; major genera were: *Staphylococcus*, *Acinetobacter*, *Pseudomonas*, *Corynebacterium*, *Klebsiella*, *Escherichia*, *Streptococcus*, *Enterococcus*, *Bacteroides* and *Proteus*. These higher rank classifications are also in agreement with the most representative species found in the data: *Staphylococcus epidermidis*, *Klebsiella pneumoniae*, *Acinetobacter baumannii* complex, *Escherichia coli*, *Staphylococcus aureus*, *Enterococcus faecalis*, *Pseudomonas aeruginosa*, *Acinetobacter iwoffi*, *Pseudomonas stutzeri* and *Staphylococcus haemolyticus*. The top 10 bacteria found for genus and species taxonomy levels in the results from ICU and NICU can be visualized in S3 Fig, faceted by time points (month) and sampling sites.

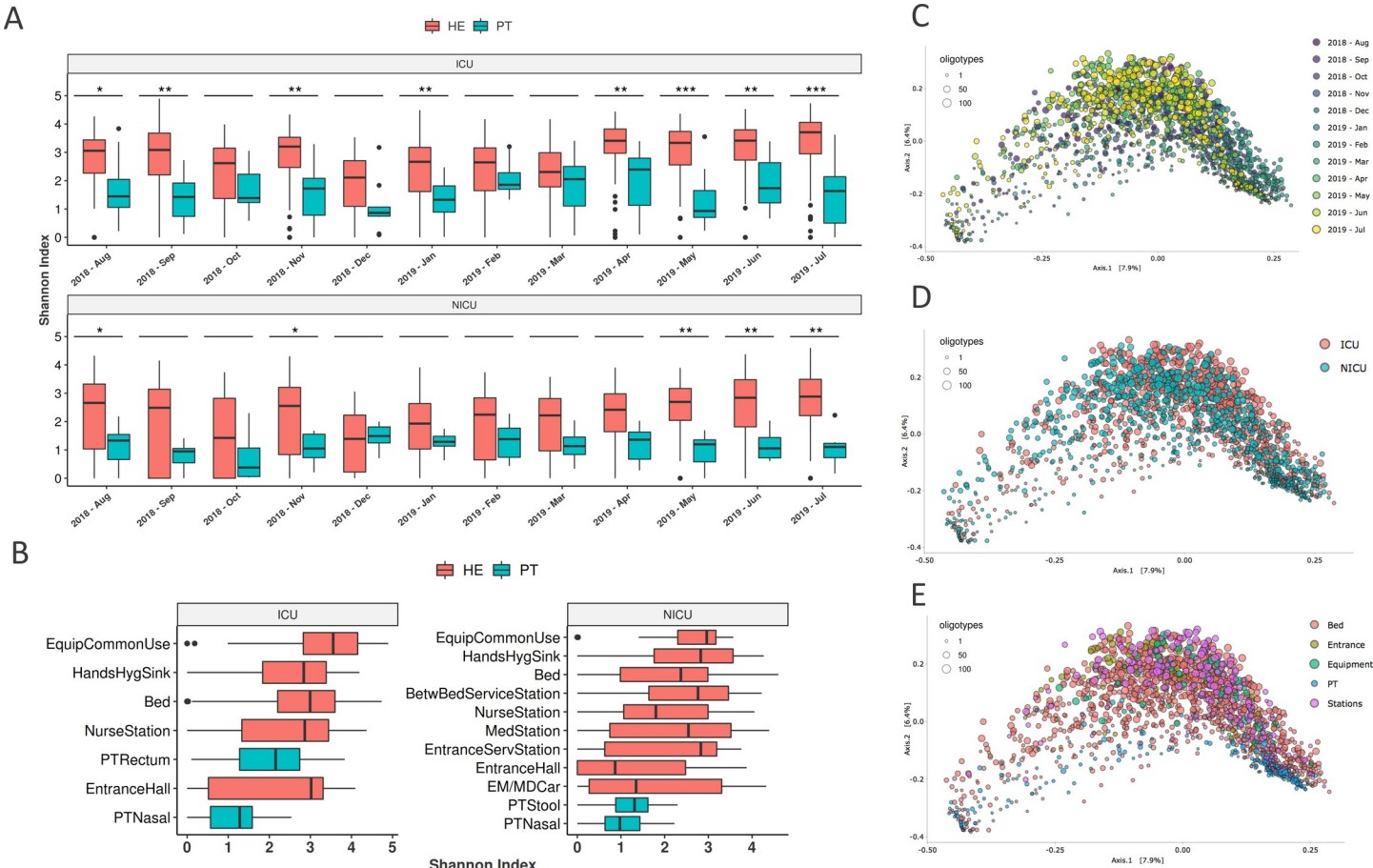

**Fig 2. Hospital diversity analysis. (A)** Shannon alpha-diversity index for ICU (top panel) and NICU (bottom panel) in healthcare environment (HE) or patient (PT) samples along the year of study. Asterisks (*) above boxplots represent the statistical significance of differences between HE and PT within each month (Wilcoxon, FDR ≤ 5%; * $p < 0.05$, ** $p < 0.01$, *** $p < 0.001$. **(B)** Shannon index calculated for ICU (left) and NICU (right) collected locations. Abbreviations: Equipment common use–EquipCommonUse, Hands Hygiene Sink–HandsHygSink, Patient Rectum–PTRectum, Patient Nasal–PTNasal, Patient Stool–PTStool, Medical Station–MedStation, Emergency/Medical car–EM/MDCar. (C-E) Beta-diversity weighted UniFrac analysis were represented by PCoA plots, considering the samples groups by **(C)** Month—August-2018 to Jul-2019, **(D)** Unit—ICU and NICU and **(E)** Collection sites: Bed, Entrance, Equipment, Patient (PT) and Stations. None of these three groups showed a clear unique diversity profile, except for patient samples more grouped in (E)–(blue–bottom right dots). Circle sizes represent the different amount of oligotypes (unique 16S sequences—sOTUs) in the samples.

Considering the previously defined HAIrB group we evaluated the proportion of positive samples for each of the individual HAIrB species in each month of the project. ICU healthcare environment and patients (Fig 3A and 3B) share high proportions of positive samples for *S. epidermidis*. In the environment, *S. epidermidis* was constant all over the year, but patient samples showed a more oscillating pattern. *A. baumannii* complex is more present in environment samples, while *E. coli* is proportionally more detected in patients' samples. In the overall profile it seems that ICU healthcare environment shows constant positivity proportions for the species across the year, while ICU patients vary for most bacteria. NICU bacteria detected in most environmental samples include *S. epidermidis*, *E. faecalis* and *A. baumanii* complex, with high proportions in samples along the year (Fig 3C). Environmental *E. coli*, *K. pneumoniae*, and *S. aureus* have proportions in specific months resembling the NICU patient positivity proportions (Fig 3D). In patient samples, the bacteria with highest detection rates were *E. faecalis*, *E. coli*, *K. pneumoniae*, and *S. epidermidis*.

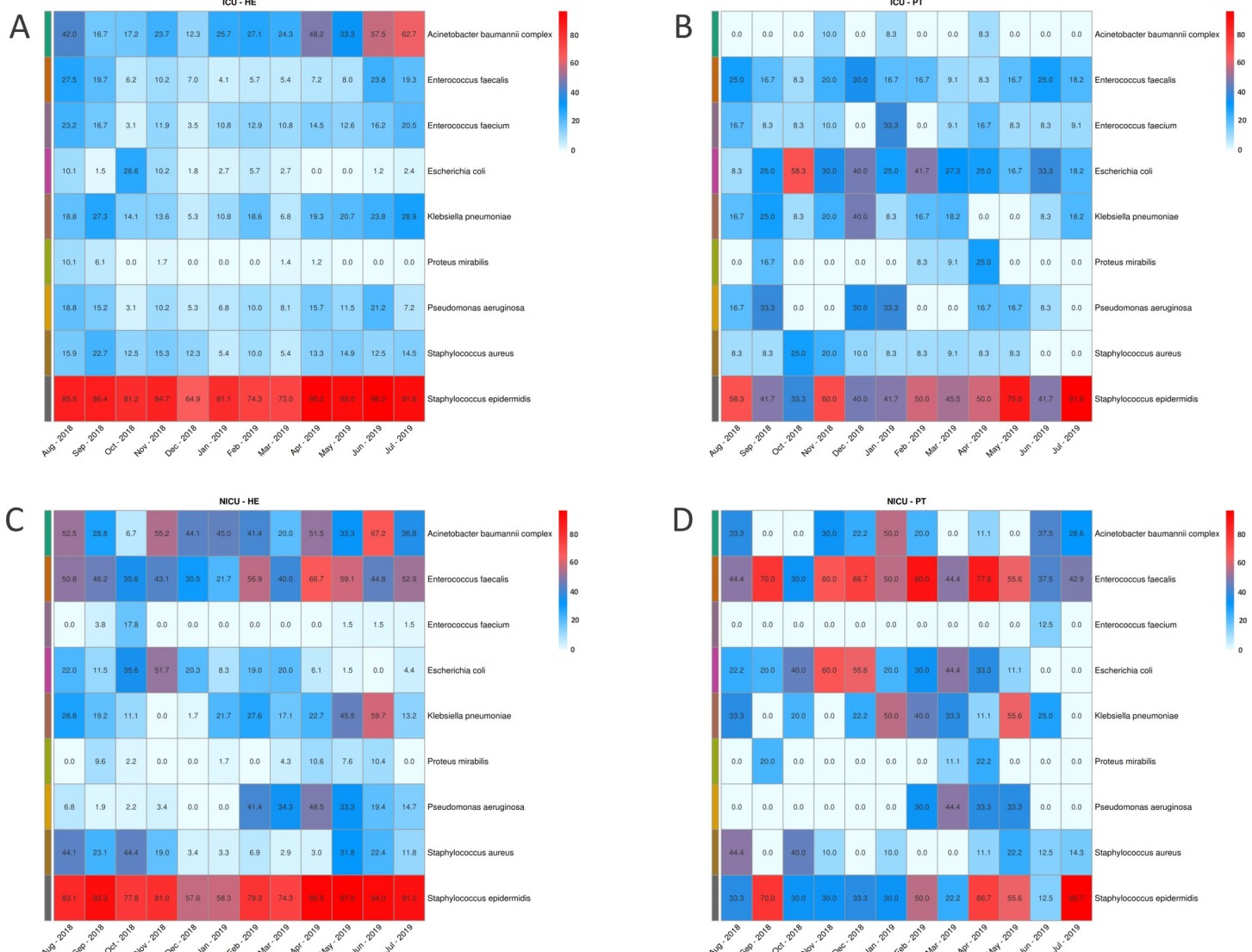

**Fig 3. Bacterial proportions detected in samples along the year.** Proportions of most abundant bacteria in samples at each month for **(A)** ICU environment, **(B)** ICU patients, **(C)** NICU environment and **(D)** NICU patients. Color scales indicate the highest proportions (in red) to the lower proportions (in white).

### Contamination hotspots and longitudinal profiles

Given the observed HAIrB annual profile, we aimed to identify possible hospital locations as contamination hotspots. Fig 4A–4F suggest that patient samples showed higher total abundance of HAIrB than environmental samples, regardless of the hospital unit. The pattern is consistent even when we split the samples according to sampling sites grouped by similarity (Fig 4B and 4E).

Within ICU, the highest HAIrB levels were observed in PT samples, with progressively lower values for bed, common use equipment, nurse station, entrance hall, and hands hygiene sink. More specifically, ICU patient rectum and nasal samples have the higher HAIrB values (Fig 4C), while the balances, taps, and alcohol dispensers showed the lowest contamination levels. NICU locations also showed highest HAIrB abundance for PT samples and lowest for entrance hall (Fig 4E). In specific sample collection sites, NICU patient stool, oximeter, and patient nasal samples have the higher HAIrB values (Fig 4F).

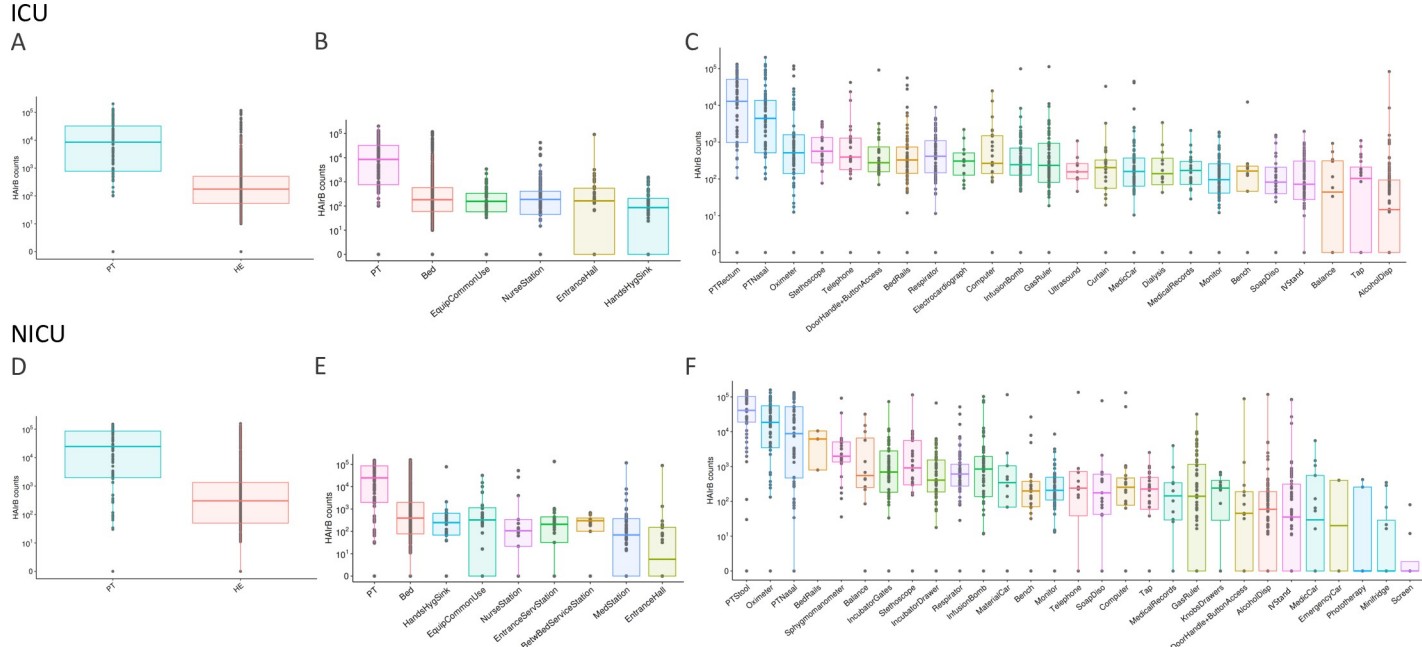

**Fig 4. Bacterial sequences detected for HAI related bacteria.** Total bacterial sequences (HAIrB counts), including *A. baumanii* complex, *E. coli*, *E. faecalis*, *E. faecium*, *K. pneumoniae*, *P. mirabilis*, *P. aeruginosa*, *S. aureus* and *S. epidermidis* in ICU samples for **(A)** patient (PT) and healthcare environment (HE), **(B)** ICU sample locations–Patient (PT), Bed, Common Use Equipments (EquipCommonUse), Nurse Station, Entrance Hall and Hands Hygiene Sinks (HandHygSink), and **(C)** specific ICU sampling sites–Patient Rectum and Nasal (PT Rectum and PT Nasal), Oximeter, Stethoscope, Telephone, Door Handle and Access buttons, Bed Rails, Respirator, Electrocardiograph, Computer, Infusion bomb, Gas Ruler, Ultrasound equipment, Curtains, Medical Car (MedCar), Dialysis, Medical records, Monitor, Bench, Soap Dispenser (SoapDisp), IV Stand, Balance, Tap and Alcohol Dispenser (AlcoholDisp). HAIrB counts for NICU samples considering **(D)** patient (PT) and healthcare environment (HE), **(E)** NICU sample locations–Patient (PT), Bed, Hands Hygiene Sinks (HandHygSink), Common Use Equipments (EquipCommonUse), Nurse Station, Entrance from service station (EntranceServStation), Service Station Between Beds (BetwBedServStation), Medical Station (MedStation) and Entrance Hall and, and **(F)** NICU specific sampling sites–Patient Stool and Nasal (PT Stool and PT Nasal), Oximeter, Bed Rails, Sphygmomanometer, Balance, Incubator gates, Stethoscope, Incubator Drawer, Respirator, Infusion Bomb, Material Car, Bench, Monitor, Telephone Soap Dispenser (SoapDisp), Computer, Tap, Medical Records, Gas Ruler, Knobs of Drawers, Door Handle and Access buttons, Alcohol Dispenser (AlcoholDisp), IV Stand, Medical Car, Emergency Car, Phototherapy, Minifridge and Srcreen.

The HAIrB profiles can be assessed with respect to their longitudinal variation (Fig 5A and 5B). While not all sites were evenly sampled, many show consistent trends. For instance, medical equipment such as oximeter, monitor, and infusion bombs showed almost constant abundance over the entire year, regardless of the hospital unit. Some sampling sites such as ICU alcohol dispenser and tap showed higher values at the end of the study period. A similar pattern was observed in NICU gas ruler, although with seemly high initial levels.

Nonetheless, many collection sites show heterogeneous profiles in terms of HAIrB contamination. For instance, NICU computer and infusion bombs, as well as ICU telephone and door handle plus access button, show well defined sample distributions outside the trend line in various months (Fig 5). This could be the result of occasional variation in decontamination practices by hospital staff, but also a sampling artifact (especially in those cases in which sample size is minimal, e.g., NICU medical records). Additionally, some sites were not evaluated at all time points. The ICU bench reached particularly low values in March-2019 and was not further monitored. Curtains and stethoscopes started being monitored near the middle of the study period. This is in agreement with the ethical guidelines imposed on this study as monitoring was primarily driven by patient-care concerns.

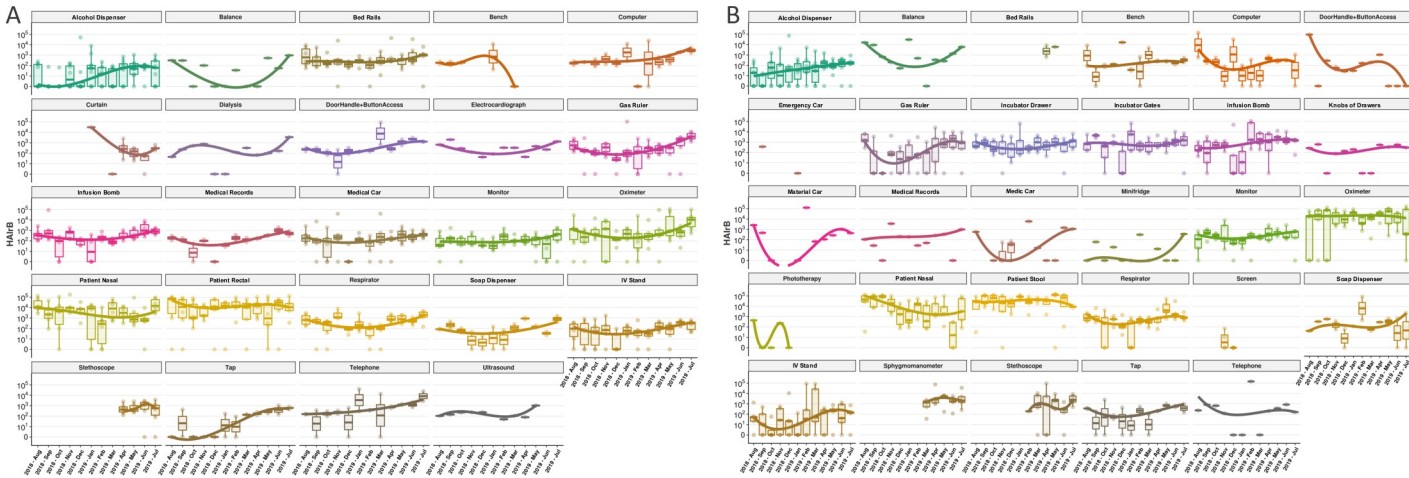

**Fig 5. Longitudinal profiles of bacteria in specific sample sites collected over the year.** (**A**) ICU and (**B**) NICU, considering the previously established HAIrB group: *A. baumanii* complex, *E. coli, E. faecalis, E. faecium, K. pneumoniae, P. mirabilis, P. aeruginosa, S. aureus* e *S. epidermidis*. HAIrB sequence reads are shown on $\log_{10}$ scale. Boxplots are ordered by median values, marked with a tendency line through the months in each sample type.

## Source tracking and bacterial dispersion

Oligotype sequences, corresponding to amplicon sequence variants (ASVs), were used to investigate the bacterial dispersion among samples and identify possible contamination flows. ICU and NICU most abundant HAIrB-related oligotypes were visualized as frequency heat-maps (Fig 6). In ICU (Fig 6A), the most frequent oligotype detected was oligotype_1, classified by our pipeline as *S. epidermidis*, and present in ~70% or more environmental samples, as well as in ~50% of patient samples. Oligotype_4, identified as *S. aureus*, was also shared across environmental and patients samples in high proportions, as well as *P. aeruginosa* oligotype_12, *K. pneumoniae* oligotypes_19, _20, _6 and _63, *E. coli* oligotype_2, *E. faecium* oligotype_24 and *E. faecalis* oligotype_5.

In general, the pointed oligotypes showed higher proportions in samples from patients and beds, regardless of the unit sampled. Still, NICU samples seem to present higher dispersion. Except for samples from the hospital entrance, all the above cited oligotypes were found in elevated proportions in patient, bed, stations, and equipment samples. *A. baumannii* complex oligotypes (_7, _3, _25 and _89) were more abundantly found in healthcare environment samples. In NICU (Fig 6B), the most frequent oligotype in all sample locations was also *S. epidermidis* oligotype_1. *E. faecalis* oligotype_5 was abundantly found in patient, bed and equipment samples, showing lower but relevant presence in stations and entrances as well. Similar to ICU oligotypes, *E. coli* oligotype_2, *P. aeruginosa* oligotype_12, *S. aureus* oligotype 4 and *K. pneumoniae* oligotypes_19, _20 and _6 were highly dispersed and shared between different location samples. In NICU, *A. baumannii* complex oligotypes were detected in both patient and environmental samples, differing from ICU where they were prevalent in environmental samples.

The longitudinal profiles of the main oligotypes and their correspondent bacteria in ICU showed that oligotype_1 appears as the main source of *S. epidermidis* throughout the year, in both environment and patients (S4A and S4B Fig). A trend line from quantile regression suggests an abundance increase of this oligotype for patients from Februrary-2019, despite lower levels in June-2019. Other ICU HAIrB did not show specific oligotype-related temporal patterns, as they presented more heterogeneous and discontinuous monthly variations (S4C–S4F Fig). NICU samples presented the same pattern for *S. epidermidis*, related to the same

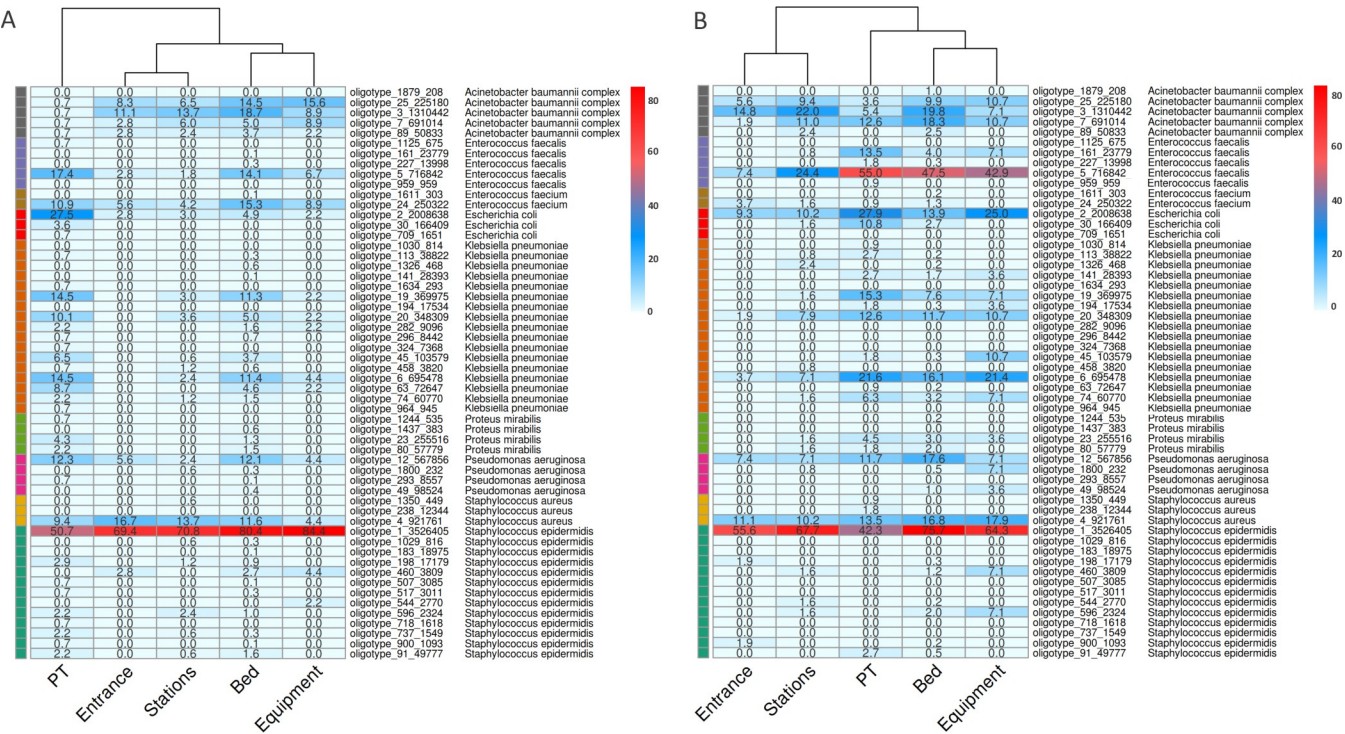

**Fig 6. Source tracking.** Heatmaps for bacterial oligotypes in (A) ICU and (B) NICU showing their proportions and distributions across the sample locations (Patients, Beds, Entrances, Equipment and Stations). Top abundant HAIrB bacteria and their specific oligotypes were selected for visualization. Oligotypes are amplicon single variants from 16S rRNA sequences, that we identified with a unique crescent code, plus its count number (Oligotype_code_count). Bacterial taxonomies, attributed by the bioinformatics pipeline, were added to the figure near the oligotype name to facilitate visualization. Color scales represent the proportion of positivity (from 80%, in red, to 0%, in white) for each oligotype in the group of samples.

oligotype_1 sequence, highly abundant across all study period in the environment (S4G and S4H Fig). Patient samples showed more significant increases from April-2019. In addition, NICU shows high abundance of *E. faecalis* oligotypes, mainly represented by oligotype_5 in patient stool samples, but also contaminating the surrounding environment—mainly the beds.

Detailed oligotype positivity rates across time points is shown in S5 Fig. In addition, these contamination frequencies for HAIrB group were plotted as risk maps for ICU and NICU (S6 and S7 Figs). These maps and red spots represent the monthly contamination in specific hospital locations, suggesting NICU has a more constantly contaminated environment than ICU, with no major HAIrB reduction over the year.

## Bacterial isolates identification and genomes characterization

ICU and NICU 17 bacterial isolates were obtained by the hospital microbiology laboratory during the same week that molecular biology samples were collected. Conventional microbiology techniques were employed as performed in the hospital routine. Then, bacterial isolates were sent to be analyzed using whole genome sequencing (WGS). The WGS results confirmed 88% of the previous microbiological results (15 out of 17 isolates) (Table 2). For the two diverging samples, one was identified just as *Klebsiella* by the microbiology lab. With WGS, we were able to classify it as *K. pneumoniae*. Another sample, identified as *K. oxytoca* by the microbiology laboratory, was classified as *K. michiganensis* by WGS analysis. Additionally, 16S rRNA gene sequencing was performed for these isolated bacteria, also resulting in 88% agreement with the conventional microbiology laboratory, differing for the same two samples of

**Table 2. Bacterial WGS sequencing.** Microorganisms isolated in the hospital microbiology laboratory in parallel to the project development (Aug-2018 to July-2019) and the subsequent additional identifications using 16S rRNA V3/V4 amplicon sequencing and whole genome sequencing (WGS) by the molecular biology laboratory.

| Month | Unit | Microbiology Lab identification | 16S rRNA -V3/V4 identity | 16S rRNA oligotype | WGS identity | MLST identity | Clonal group identity |
|---|---|---|---|---|---|---|---|
| N/A | N/A | *Staphylococcus aureus* | *Staphylococcus aureus* | oligotype_4 | *Staphylococcus aureus* | ST1435 | N/D |
| Oct-18 | NICU | *Staphylococcus aureus* | *Staphylococcus aureus* | oligotype_4 | *Staphylococcus aureus* | ST2383 | *S. aureus* Group 1 |
| Oct-18 | NICU | *Staphylococcus aureus* | *Staphylococcus aureus* | oligotype_4 | *Staphylococcus aureus* | ST398 | *S. aureus* Group 2 |
| Oct-18 | NICU | *Staphylococcus aureus* | *Staphylococcus aureus* | oligotype_4 | *Staphylococcus aureus* | ST2383 | *S. aureus* Group 1 |
| Jan-19 | NICU | *Staphylococcus aureus* | *Staphylococcus aureus* | oligotype_4 | *Staphylococcus aureus* | ST398 | *S. aureus* Group 2 |
| Nov-18 | ICU | *Staphylococcus epidermidis* | *Staphylococcus epidermidis* | oligotype_1 | *Staphylococcus epidermidis* | ST2 | N/D |
| Sep-18 | ICU | *Enterococcus faecalis* | *Enterococcus faecalis* | oligotype_5 | *Enterococcus faecalis* | ST21 | N/D |
| Feb-19 | NICU | *Enterococcus faecalis* | *Enterococcus faecalis* | oligotype_5 | *Enterococcus faecalis* | ST21 | N/D |
| Dec-18 | ICU | Pseudomonas aeruginosa | Pseudomonas aeruginosa | oligotype_12 | *Pseudomonas aeruginosa* | ST245 | N/D |
| Feb-19 | NICU | Pseudomonas aeruginosa | Pseudomonas aeruginosa | oligotype_12 | *Pseudomonas aeruginosa* | ST1816[1] | N/D |
| Oct-18 | ICU | *Klebsiella* | *Klebsiella pneumoniae* | N/D | *Klebsiella pneumoniae* | ST307 | N/D |
| Oct-18 | NICU | *Klebsiella oxytoca* | *Enterobacteriaceae* | N/D | *Klebsiella michiganensis* | N/A | N/D |
| Nov-18 | ICU | *Escherichia coli* | *Escherichia coli* | oligotype_2 | *Escherichia coli* | ST131 | N/D |
| Jan-19 | NICU | *Escherichia coli* | *Escherichia coli* | N/D | *Escherichia coli* | ST69 | *E. coli* Group 1 |
| Jan-19 | NICU | *Escherichia coli* | *Escherichia coli* | N/D | *Escherichia coli* | ST69 | *E. coli* Group 1 |
| Oct-18 | ICU | *Escherichia coli* | *Escherichia coli* | oligotype_2 | *Escherichia coli* | ST410 | N/D |
| Sep-18 | ICU | *Escherichia coli* | *Escherichia coli* | oligotype_2 | *Escherichia coli* | ST10 | N/D |

[1] Has one allele with less than 100% identity (99.7992%).

N/D–Not identified.

N/A–Not applicable. Sample was collected before the actual project starts.

*Klebsiella*. One of them (*Klebsiella*) was classified as *K. pneumoniae* by our bioinformatics pipeline and the other one (*K. oxytoca*) could not be differentiated below Enterobacteriaceae family level. Thus, WGS and 16S rRNA sequencing presented 94,11% agreement in the taxonomical classifications, only differing in the resolution level for the *K. michiganensis* in the amplicon 16S rRNA gene sequencing (Table 2).

Genomic investigation analysis was performed by searching for the specific 16S rRNA gene sequences in each isolated WGS results and comparing them to the oligotypes identified in the ICU and NICU amplicon samples. The same 16S rRNA fragment sequence, corresponding to oligotype_4, was found in all *S. aureus* isolated genomes—including the one from before the project period. Two *E. coli* bacterial isolates from NICU in 2019 did not have any correspondence for their 16S rRNA sequences in the amplicon hospital microbiome survey. Furthermore, specific 16S rRNA sequences from *Klebsiella* WGS results, isolated from ICU and NICU, were not detected in the hospital microbiome oligotypes either.

In addition, genomic MLST profiling was performed along with clonality analysis using WGS results. These analyses separated *S. aureus* in three MLST strains (ST398, ST2383 and ST1435) and two clonal groups (*S. aureus* Group 1 and 2). Group 2 includes bacterial isolates from October-2018 and January-2019, both from NICU, however, these two isolates were more distantly related (195 SNVs, 99.93 ANIm and an average of 98.47% nucleotides aligned) than the Group1 isolates from October-2018 (97 SNVs, 99.99 ANIm and an average of 99.90% nucleotides aligned). The most different *S. aureus* (ST1435—isolated in the hospital prior to this project) was the only one carrying a *mecA* resistance gene. In addition, the *S. epidermidis* isolated from the ICU in November-2018 belongs to ST2 and also carries the *mecA* resistance

gene. The two *E. faecalis* were from the same MLST strain (ST21), but not sufficiently similar to be considered clonal (>5,000 SNVs, 99.59 ANIm and an average of 92.18% nucleotides aligned). One *E. faecalis* was isolated from ICU and the other from NICU. *P. aeruginosa* isolates, both from ICU, were identified as different MLST strains (ST245 and ST1816* containing 1 SNP from the canonical strain) and not clonally related (>9,000 SNVs, 99.29 ANIm and an average of 90.29% nucleotides aligned). Only two *E. coli* isolated from NICU in January-2019 belongs to the same MLST strain (ST69) and clonal group (*E. coli* Group 1—with 10 SNVs, 99.97 ANIm and an average of 99.74% nucleotides aligned). However, these clonal *E. coli* genomes are the ones without corresponding 16S rRNA oligotypes in the microbiome survey. This WGS analysis from bacterial isolates corroborates our bioinformatics pipeline taxonomical identification for the species-level classification of most relevant oligotypes identified in this microbiome project.

## Antimicrobial resistance genes profile in the hospital microbiome samples

Antimicrobial resistance genes were searched monthly in healthcare environment and patient samples in both ICU and NICU. In ICU, a total of 135 environmental samples (14%) and 51 patient samples (36%) were positive for any resistance gene. In 3.7% of environmental samples and 15% of patient samples more than one resistance gene was detected. In 72 ICU beds tested, 22 of them showed the same resistance gene found in samples from its respective inpatient. Resistance genes positivity over the twelve-month period in the ICU showed that *mecA*, $bla_{\text{CTX-M-1 group}}$, $bla_{\text{SHV-like}}$, $bla_{\text{KPC-like}}$ and *vanA* were the most prevalent genes detected (Fig 7A). Other genes were detected in lower frequencies and in different periods, such as $bla_{\text{SPM-like}}$ detected from May to June– 2019, $bla_{\text{OXA-51-like}}$ detected from January to May– 2019, and $bla_{\text{NDM-like}}$ not continuously detected. Patients showed more positive samples than healthcare environment (Fig 7B). However, proportionally, patients have far fewer samples collected than environment. In terms of specific sampling sites (Fig 7C), *mecA* gene was found widely distributed and with higher proportions in samples such as patient nose and equipment as telephone, door handles/access buttons, respirators, computers, electrocardiograph, gas ruler, oximeters, among others. $bla_{\text{KPC-like}}$ was frequently found in dialysis equipment, stethoscope, gas ruler and bedrails, as well as in patient rectum samples and similar to $bla_{\text{SHV-like}}$ distributions. *vanA* gene distributions is also higher in stethoscopes, electrocardiograph, bedrails, infusion bomb, and patient rectum samples.

In NICU, a total of 112 environmental samples (13%) and 32 patient samples (30%) tested positive for any resistance gene. In 1.6% of environmental samples and 9.9% patient samples more than one resistance gene was detected. Among 60 NICU tested beds, 19 showed the same resistance genes as its respective inpatient. Assessment of AMR genes in the NICU demonstrated higher positivity of *mecA* gene over all the twelve months, with higher frequencies from August-2018 to November-2018 and from May-2019 to July-2019 (Fig 7D). $bla_{\text{CTX-M-1 group}}$, $bla_{\text{SHV-like}}$ were also detected, but most frequently from January-2019 to June-2019. Patients also showed high positivity for AMR genes, with exception for *mecA* gene, which was even more present in environmental samples (Fig 7E). Also, *mecA* gene was found in higher proportions of most samples analyzed (Fig 7F), but even more frequent in respirators, sphygmomanometers, material cars, incubators gates and drawers, gas ruler, and stethoscopes samples. $bla_{\text{SHV-like}}$ was mostly found in patient stool samples as well as in balance, bedrail and sphygmomanometer samples.

Correlating the most abundant AMR genes in patients with healthcare environments, and adjusting the proportion of samples for each group, we could observe that there is an overlapping frequency pattern: when the AMR gene is in higher frequencies for environmental

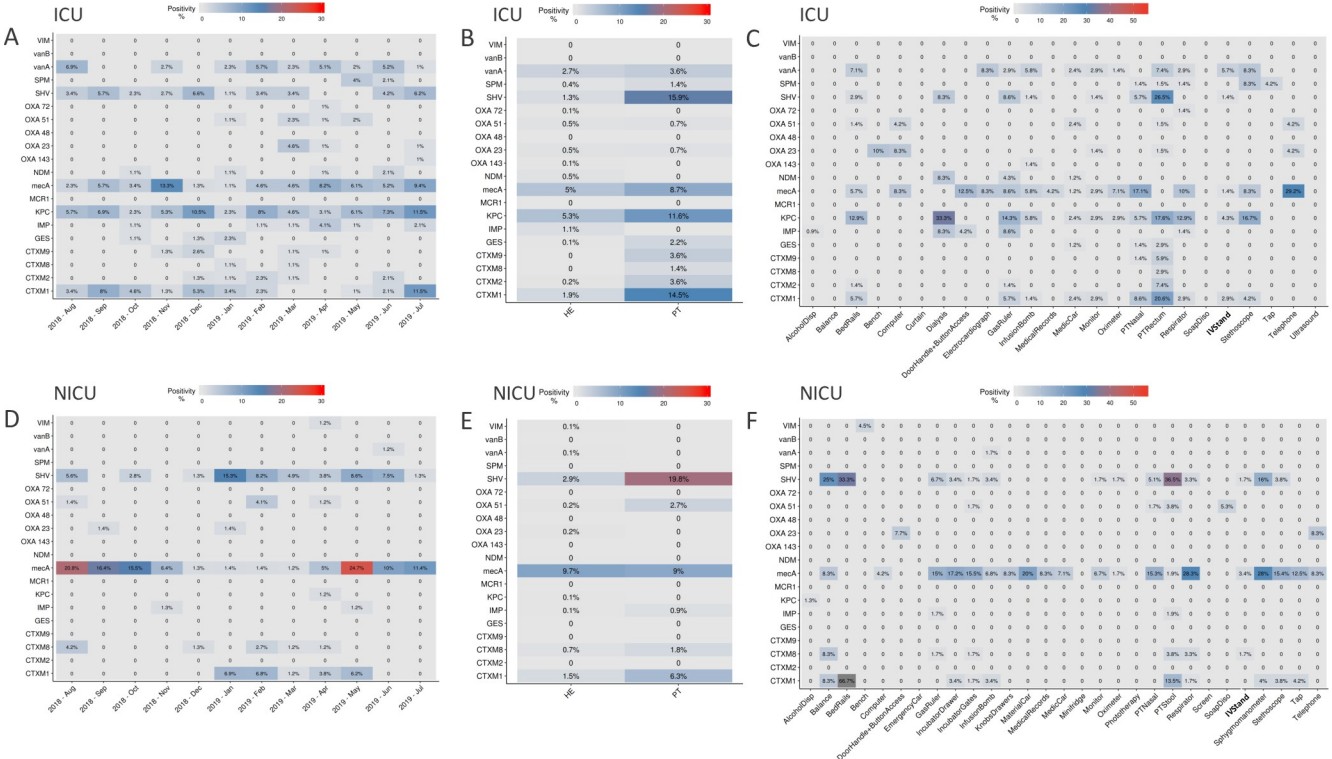

**Fig 7. Antimicrobial resistance genes proportions of positivity in samples.** (**A**) AMR genes in the ICU over the year, (**B**) in the ICU between sources—patient (PT) and healthcare environment (HE), (**C**) in the ICU among specific collection sites: Alcohol Dispensers (AlcoholDisp), Balance, Bed Rails, Bench, Computer, Curtains, Dialysis, Door Handle and Access Buttons, Electrocardiograph, Gas Ruler, Infusion Bomb, Medical Records, Medic Car, Monitor, Oximeter, Patient Nasal and Rectum (PTNasal and PTRectum), Respirator, Soap Dispenser (SoapDisp), IV Stand, Stethoscope, Tap, Telephone, Ultrasound.(**D**) Resistance genes in the NICU along the year, (**E**) in the NICU between sources—patient (PT) and environment (HE) and (**F**) in the NICU among specific sample collection sites: Alcohol Dispensers (AlcoholDisp), Balance, Bed Rails, Bench, Computer, Door Handle and Access Buttons, Emergency car, Gas Ruler, Incubator Drawer, Material Car, Medical Records, Medic Car, Minifridge, Monitor, Oximeter, Phototherapy, Patient Nasal and Stool (PTNasal and PTStool), Respirator, Screen, Soap Dispenser (SoapDisp), IV Stand, Sphymomanometer, Stethoscope, Tap and Telephone.

samples, they are also higher in patient samples (Fig 8). Given the experimental design carried out in this study, we were not able to define the actual contamination source, the environment or the patient. We could, nonetheless, observe their close relationship.

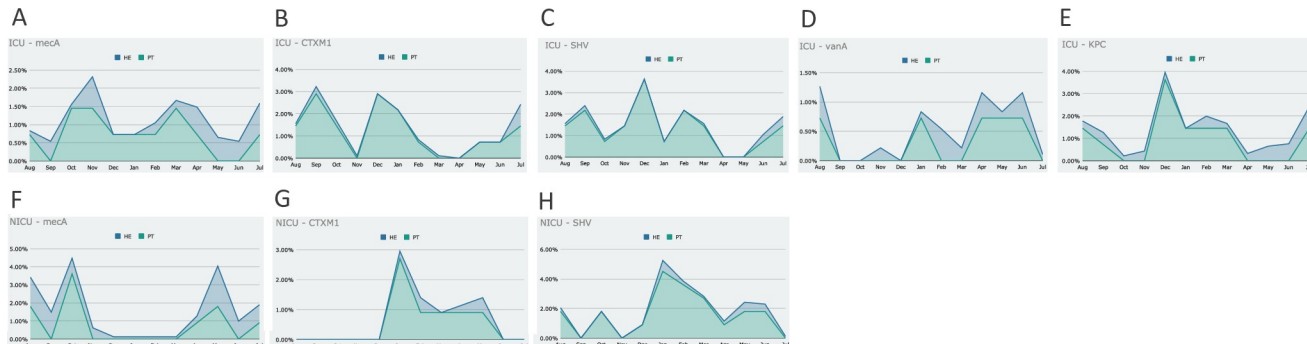

**Fig 8. Antimicrobial resistance gene frequencies in patients (PT) and healthcare environment (HE) for each month.** (**A**) *mecA* gene in ICU, (**B**) $bla_{CTX-M-1}$ group gene in ICU, (**C**) $bla_{SHV-like}$ gene in ICU, (**D**) *vanA* gene in ICU, (**E**) $bla_{KPC-like}$ gene in ICU, (**F**) *mecA* gene in NICU, (**G**) $bla_{CTX-M-1}$ group gene in NICU, (**H**) $bla_{SHV-like}$ gene in NICU.

## Discussion

Hospital microbiome profiling, and more specifically bacterial pathogen tracking, can be of great relevance to understand, reduce and prevent HAI. In this paper we have demonstrated the application of NGS amplicon sequencing to screen, monitor and track bacterial profiles in a hospital environment. Additionally, we used the same NGS samples for AMR/MDR genes profiling using real-time PCR and compared the amplicon results with WGS analysis of bacterial isolates from ICU and NICU.

Several bacterial investigations have been carried out in intensive care units, since ICUs are related to high rates of HAI and MDR acquired infections [4,17,45–47]. Some ICU studies using high-throughput sequencing reveal bacterial profiles that would not be recovered only by conventional microbiology methods, given its specificity and selectivity for known types of bacteria [5,48,49]. The method we used in this study was also culture-independent but designed to allow comparisons relative to the bacterial load in the samples. Previous studies have already demonstrated that total sequence reads from NGS samples (library sizes) do not need to be arbitrary, thereby allowing bacterial load estimation [28,50]. This allows temporal tracking of contamination profiles which can be critical to guide hospital infection control practices.

The bacterial profile found in this study for ICU and NICU environments is similar to other intensive care studies, with Proteobacteria and Firmicutes as the most abundant bacterial phyla detected [1,2,5]. However, most abundant bacterial genera and species detected in our study varied, particularly because of the high rates of specific HAI related bacteria.

The overall results show that the hospital environment maintains a close relationship with the patients' microbiome in their respective care units, a pattern also observed for AMR genes. In this project, the experimental design was not constructed to infer the definitive sources of bacterial contaminations (patients, environment, or healthcare professionals). Such inferences have been addressed by previous studies which were able to associate patient infection and environmental contamination while accounting for duration of patient stay within the hospital [1]. However, we observed the NICU bacterial profiles with wide and continuous contaminations over the year, suggesting a flow between patients and environment by factors like healthcare workers or visitors. Other studies already showed the resemblance between NICU environment and the gut microbes of premature infants [51]. We observed some positivity rate reduction in NICU for *S. aureus*, *S. epidermidis* and *mecA* AMR gene during November and December of 2018, that might be attributed to infection control interventions in this period. In the ICU, particularly rigorous hygienization processes were also reported by the hospital in December-2018. This could explain the variations in the bacterial and AMR profiles during this period and shortly after.

In both ICU and NICU, throughout our study period, patient samples yielded the highest bacterial load, despite their lower alpha-diversity indexes. Yet, patients were the only sample group with apparent clustering in beta-diversity analysis. This patient profile can be explained by the fact that they showed a particular microbiota, with predominance of only a few bacteria, which is generally expected for nasal samples but not for fecal/rectal samples [52,53]. This diversity may be related to patient infections or microbiota imbalance due to hospitalization [54–56]. Recent studies have shown that newborns are colonized by the maternal and surrounding microbiota immediately after birth. Infants from cesarean sections, mothers treated with prophylactic antibiotics, or not breastfed during the neonatal period have altered microbiota profiles, as well as colonization by opportunistic pathogens associated with the hospital environment, such as *Enterococcus*, *Enterobacter* and *Klebsiella* (Shao et al., 2019). The implications of microbiota changes during hospitalization period is not fully understood. Another healthcare issue that may be directly associated with microbiome diversity is the development

of sepsis. Studies have identified low intestinal microbiota diversity and *Staphylococcus* predominance as risk factors for sepsis in neonates born at 24–27 weeks of gestation [57]. In adults, the risk of sepsis has also been studied related to the microbiome profile imbalance and other associated factors [58].

We detected AMR genes more frequently in rectal/stool samples, but also with considerable proportions in nasal samples. *mecA*, *bla*$_{CTX-M-1\ group}$, *bla*$_{SHV-like}$ and *bla*$_{KPC-like}$ were by far the most frequent AMR genes detected in patients, but also spread in the hospital environment. Meticillin-resistant *S. aureus* (MRSA) carrying *mecA* gene is a rising threat to public health [59]. This gene has been found in hospital environments and circulating in the community. Also, *mecA* gene is not only restricted to *S. aureus*, but also detected among other species from the genera, including *S. epidermidis* [60,61]. In our study, patient nasal samples from both ICU and NICU were predominant for *mecA* AMR gene, which could be related to most abundant gram-positive bacteria found in nasal samples: *S. epidermidis*. Carbapenem resistance and Extended spectrum β-lactamases genes detected in this study (*bla*$_{KPC-like}$, *bla*$_{CTX-M-1\ group}$, *bla*$_{SHV-like}$ and) were commonly reported in other studies from Brazilian hospital environments as a serious problem in urgent need of actions to reduce spreading [62,63]. In our study, higher frequencies of these Carbapenemases and β-lactamases were found in patient rectum/stool samples, which are also low diversity samples with predominance of gram-negative bacteria like *Escherichia* and *Klebsiella*.

In both NICU and ICU, we highlight the large bacterial variation in the collection sites. In each month, there were samples indicating divergent degrees of contamination within each collection sample group. This suggests a lack of homogeneity or reproducibility in the sanitation processes for similar samples, or the rapid re-contamination of the environment after cleaning. This study was not designed to focus on the evaluation of the sanitation processes, but rather to deepen our understanding of the hospital bacteriome. WGS analysis of bacterial isolates allowed us to complement the amplicon sequencing results by confirming the taxonomical identification for several 16S rRNA amplicons and by showing that bacterial microbiome profiles, recovered in patient and environmental samples, were from significant bacterial strains widely spread in the hospital intensive care units.

Therefore, through a one-year-long hospital surveillance study we were able to demonstrate how NGS technologies can be applied to large-scale monitoring of hospital microbial contamination, thereby improving infection control practices. Effective identification of HAI hotspots and contamination flows with pathogenic bacteria, in multiple sampling sites simultaneously, represents an unprecedented resolution gain for hospital microbiological control and epidemiological surveillance.

## Supporting information

**S1 Fig. Shannon diversity profiles.**
(PDF)

**S2 Fig. Weighted UniFrac beta-diversity profiles for all samples collected in both units, ICU and NICU separated in three different timelines.**
(PDF)

**S3 Fig. Most abundant bacteria detected and classified in genus and species.**
(PDF)

**S4 Fig. Longitudinal profile of specific bacteria and its oligotypes most abundantly found in ICU and NICU.**
(PDF)

**S5 Fig. Proportion of positive samples for the most abundant oligotypes in ICU and NICU.**
(PDF)

**S6 Fig. Risk map from the ICU along the year.**
(PDF)

**S7 Fig. Risk map from the NICU along the year.**
(PDF)

**S1 Table. Detailed sample collection sites in the project.**
(PDF)

**S2 Table. Monthly collected samples and sequencing information.**
(PDF)

## Acknowledgments

We thank Prof. Dr. Ana Cristina Gales (ALERTA Laboratory, Federal University of São Paulo–USP) who kindly provided positive control strains for the resistance gene (RGene) study.

## Author Contributions

**Conceptualization:** Aline Fernanda Rodrigues Sereia, Camila Hernandes, Telma Priscila Lovizio Raduan, Marines Dalla Valle Martino, Fernando Gatti de Menezes, Luiz Felipe Valter de Oliveira.

**Data curation:** Ana Paula Christoff, Aline Fernanda Rodrigues Sereia, Giuliano Netto Flores Cruz, Vanessa Leitner Silva, Luiz Felipe Valter de Oliveira.

**Formal analysis:** Ana Paula Christoff, Aline Fernanda Rodrigues Sereia, Giuliano Netto Flores Cruz, Daniela Carolina de Bastiani, Vanessa Leitner Silva, Luiz Felipe Valter de Oliveira.

**Investigation:** Daniela Carolina de Bastiani, Vanessa Leitner Silva, Ana Paula Metran Nascente, Ana Andrea dos Reis, Renata Gonçalves Viessi, Andrea dos Santos Pereira Marques, Bianca Silva Braga, Telma Priscila Lovizio Raduan.

**Methodology:** Ana Paula Christoff, Aline Fernanda Rodrigues Sereia, Telma Priscila Lovizio Raduan, Marines Dalla Valle Martino, Luiz Felipe Valter de Oliveira.

**Project administration:** Aline Fernanda Rodrigues Sereia, Camila Hernandes, Telma Priscila Lovizio Raduan, Marines Dalla Valle Martino, Fernando Gatti de Menezes, Luiz Felipe Valter de Oliveira.

**Supervision:** Aline Fernanda Rodrigues Sereia, Camila Hernandes, Telma Priscila Lovizio Raduan, Luiz Felipe Valter de Oliveira.

**Validation:** Giuliano Netto Flores Cruz.

**Writing – original draft:** Ana Paula Christoff.

**Writing – review & editing:** Ana Paula Christoff, Aline Fernanda Rodrigues Sereia, Giuliano Netto Flores Cruz, Luiz Felipe Valter de Oliveira.

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
