## [Decision Letter · Decision Letter 0]

22 Apr 2020

PONE-D-20-08031

One year cross-sectional study in adult and neonatal intensive care units reveals the bacterial and antimicrobial resistance genes profiles in patients and hospital surfaces

PLOS ONE

Dear Dr. Oliveira,

Thank you for submitting your manuscript to PLOS ONE. After careful consideration, we feel that it has merit but does not fully meet PLOS ONE’s publication criteria as it currently stands. Therefore, we invite you to submit a revised version of the manuscript that addresses the points raised during the review process.

We would appreciate receiving your revised manuscript by Jun 06 2020 11:59PM. To enhance the reproducibility of your results, we recommend that if applicable you deposit your laboratory protocols in protocols.io, where a protocol can be assigned its own identifier (DOI) such that it can be cited independently in the future. For instructions see: http://journals.plos.org/plosone/s/submission-guidelines#loc-laboratory-protocols

We look forward to receiving your revised manuscript.

Kind regards,

Zhi Ruan, Ph.D.

Academic Editor

PLOS ONE

"I have read the journal's policy and the authors of this manuscript have the following competing interests: AC, AS, GC, DB, VS, LO are currently full-time employees of BiomeHub (SC, Brazil), a research and consulting company specialized in microbiome technologies."

Reviewers' comments:

Reviewer's Responses to Questions

**Comments to the Author**

1. Is the manuscript technically sound, and do the data support the conclusions?

Reviewer #1: Yes

Reviewer #2: Yes

2. Has the statistical analysis been performed appropriately and rigorously? 

Reviewer #1: No

Reviewer #2: Yes

3. Have the authors made all data underlying the findings in their manuscript fully available?

Reviewer #1: No

Reviewer #2: Yes

4. Is the manuscript presented in an intelligible fashion and written in standard English?

Reviewer #1: Yes

Reviewer #2: Yes

5. Review Comments to the Author

Reviewer #1: Overall comments: The English language could be reviewed and improved throughout the document, mainly with some sentence constructions.

The figure legends as well as legends for supplementary files need to be more detailed, referring to various aspects of the figure components.

Line 79-82: I think this needs rewording, the message in the sentence seems to be lost.

Lines 202-203: What were the positive controls used for the RT-PCR? And probe details? This information could be useful and should be included in the supplementary information.

Line 209: What was the criteria used to select isolates for genome sequencing? This information needs to be elaborated in the methods.

Lines 252-255: Could the statistical significance for these differences be indicated in the figure and explained in the figure legend?

Lines 278-279: (Supp Fig 3) I appreciate that this is only the supplementary figure, but it is nevertheless almost incomprehensible. Maybe stick to only genus and species panels?

Line 438: In Figure 7, it would be good if the ICU and NICU panels could be indicated for clarity in the figure

Lines 529-531: These sentences need to be reworded.

Line 557: A search shows the BioProject data to be currently unavailable. This needs to be addressed before this manuscriot can be accepted.

Reviewer #2: Thank you for allowing me to review this very interesting and timely paper, One year cross-sectional study in adult and neonatal intensive care units reveals the bacterial and antimicrobial resistance genes profiles in patients and hospital surfaces. It is an important topic and this reviewer has a few comments. Overall, the paper is written clearly. The introduction could provide the reader with more background on the infection issues in critical care units and mention the difference observed between the neonatal and adult ICUs. The authors need to provide a reason and justification why they collected rectal swabs on adults and stool specimens on infants. These are two different microbiomes that the authors are attempting to compare. The authors need to provide the reader a summary of the neonatal and adult units that were the sites of the study-type of patients, tertiary hospital, etc.

There are many figures in the main paper and many supplementary tables. Supplementary table 1 is very valuable in that it tells the reader the location of samples and the number of samples obtained from that area. This is an important table and perhaps needs to be in the main body of the paper. Overall the figure and table legends need more information and need to define each abbreviation for the reader to interpret the figure/ table. For example in Table 1 first row, it is not clear what the purpose of this entry (*Isolated in the hospital previously the project period). Figure 2 is a great figure but it was difficult to decipher the color legends in C, D, and E. Again, the maps of the units (Supp. Figures 7&8) contain very valuable information and may help explain why some locations in the units are more contaminated than other however, the figure are difficult to read.

The discussion mentions the nasal samples and the resistance genes that were identified but there is little mention in the paper of a comparison of nasal versus gut microbiome of the patients-resistant genes versus bacteria. A limitation paragraph might be helpful at the end of the discussion.

Please double-space the manuscript.

6. PLOS authors have the option to publish the peer review history of their article (what does this mean?). If published, this will include your full peer review and any attached files.

Reviewer #1: No

Reviewer #2: No

---

## [Author Response · Author response to Decision Letter 0]

1 May 2020

RESPONSE: We revised the style requirements and necessary corrections were made according to PLOS ONE manuscript body formatting guidelines (February 2020).

I have read the journal's policy and the authors of this manuscript have the following competing interests: AC, AS, GC, DB, VS, LO are currently full-time employees of BiomeHub (SC, Brazil), a research and consulting company specialized in microbiome technologies. 

RESPONSE: We confirmed that the competing interests does not alter our adherence to all PLOS ONE policies and performed the requested updates in the cover letter with the following statement - I have read the journal's policy and the authors of this manuscript have the following competing interests: AC, AS, GC, DB, VS, LO are currently full-time employees of BiomeHub (SC, Brazil), a research and consulting company specialized in microbiome technologies. This does not alter our adherence to PLOS ONE policies on sharing data and materials. All authors have read and approved the manuscript.

Response to reviewers

We thank to the reviewers for the significative contributions in revising this manuscript. All the questions and suggestions helped to improve the manuscript quality. Responses for each point were addressed below.

Reviewer #1: Overall comments: The English language could be reviewed and improved throughout the document, mainly with some sentence constructions.

RESPONSE: The manuscript was revised and sentences rephrased to improve the English language.

The figure legends as well as legends for supplementary files need to be more detailed, referring to various aspects of the figure components.

RESPONSE: All figure legends were revised and improved.

Line 79-82: I think this needs rewording, the message in the sentence seems to be lost.

RESPONSE: The sentence was revised and rewritten. 

Lines 202-203: What were the positive controls used for the RT-PCR? And probe details? This information could be useful and should be included in the supplementary information.

RESPONSE: In fact, there is no RT-PCR assay in the manuscript. The Real-Time PCR was performed based on bacterial genomic DNA, without reverse transcription (RT). Probe details were already included in the methods section. We added an statement about positive controls, along with an acknowledgement for the Prof. Dr. Ana Gales, that kindly give us the positive controls strains.

Line 209: What was the criteria used to select isolates for genome sequencing? This information needs to be elaborated in the methods.

RESPONSE: We included an statement in the methods that bacterial isolates from the hospital routine surveillance swabs, during the same week that the molecular biology collection took place, were selected for whole genome sequencing.

Lines 252-255: Could the statistical significance for these differences be indicated in the figure and explained in the figure legend?

RESPONSE: Wilcoxon statistical test was performed and included in the figure. The statistical differences are indicated in the figure legend.

Lines 278-279: (Supp Fig 3) I appreciate that this is only the supplementary figure, but it is nevertheless almost incomprehensible. Maybe stick to only genus and species panels?

RESPONSE: Fig S3 was changed according to the suggestions.

Line 438: In Figure 7, it would be good if the ICU and NICU panels could be indicated for clarity in the figure

RESPONSE: Panels were indicated for ICU and NICU in Fig 7.

Lines 529-531: These sentences need to be reworded.

RESPONSE: The sentences were rewritten.

Line 557: A search shows the BioProject data to be currently unavailable. This needs to be addressed before this manuscriot can be accepted.

RESPONSE: BioProject PRJNA604445 was made available in the NCBI SRA repository https://www.ncbi.nlm.nih.gov/bioproject/PRJNA604445

Reviewer #2: Thank you for allowing me to review this very interesting and timely paper, One year cross-sectional study in adult and neonatal intensive care units reveals the bacterial and antimicrobial resistance genes profiles in patients and hospital surfaces. It is an important topic and this reviewer has a few comments. 

Overall, the paper is written clearly. The introduction could provide the reader with more background on the infection issues in critical care units and mention the difference observed between the neonatal and adult ICUs. 

RESPONSE: A new paragraph was included in the introduction section to clarify this.

The authors need to provide a reason and justification why they collected rectal swabs on adults and stool specimens on infants. These are two different microbiomes that the authors are attempting to compare.

RESPONSE: This experimental design was restricted by the hospital ethics committee, which didn’t allowed the collection of rectal swabs from the neonates in this project. Then we performed the best that we could do for obtaining the results, and defined the collection of freshly emitted stool from neonates. We now added this information on fresh stool in the methods section.

The authors need to provide the reader a summary of the neonatal and adult units that were the sites of the study-type of patients, tertiary hospital, etc.

RESPONSE: The information about the hospital (tertiary and public) were reinforced along the manuscript. There was only one ICU and one NICU unit in the hospital, the collection sites were the same for each month. Detailed sample collection sites were depicted in the S2 Table.

There are many figures in the main paper and many supplementary tables. Supplementary table 1 is very valuable in that it tells the reader the location of samples and the number of samples obtained from that area. This is an important table and perhaps needs to be in the main body of the paper. 

RESPONSE: We included a table 1 in the methods section with a resumed view of the collection sites and number of samples.

Overall the figure and table legends need more information and need to define each abbreviation for the reader to interpret the figure/ table. For example in Table 1 first row, it is not clear what the purpose of this entry (*Isolated in the hospital previously the project period). Figure 2 is a great figure but it was difficult to decipher the color legends in C, D, and E. Again, the maps of the units (Supp. Figures 7&8) contain very valuable information and may help explain why some locations in the units are more contaminated than other however, the figure are difficult to read.

RESPONSE: All figure legends were modified and improved. Table 1 legend was altered, and the dubious information explained in the text. Figure 2 legend was fixed to improve comprehension of the colors and collection sites evaluated in each PCoA plot. In supplementary S7 and S8 figures the figure legends were improved but the main purpose of the maps is to give a wide view of the contamination status over the year, in a timely way, where we can see that in ICU the hotspots were in fewer locations varying each month but in NICU, they were constant in pretty much every month and location. It will be too complex to associate specific location for hotspots and the hospital blueprints in that figures. If they still difficult we can remove them.

The discussion mentions the nasal samples and the resistance genes that were identified but there is little mention in the paper of a comparison of nasal versus gut microbiome of the patients-resistant genes versus bacteria. A limitation paragraph might be helpful at the end of the discussion.

RESPONSE: The discussion was improved and such comparisons were included in the text.

Please double-space the manuscript.

RESPONSE: It was corrected along with other manuscript formatting requested by the journal guidelines.

---

## [Decision Letter · Decision Letter 1]

20 May 2020

One year cross-sectional study in adult and neonatal intensive care units reveals the bacterial and antimicrobial resistance genes profiles in patients and hospital surfaces

PONE-D-20-08031R1

Dear Dr. Oliveira,

We are pleased to inform you that your manuscript has been judged scientifically suitable for publication and will be formally accepted for publication once it complies with all outstanding technical requirements.

With kind regards,

Zhi Ruan, Ph.D.

Academic Editor

PLOS ONE

Reviewers' comments:

Reviewer's Responses to Questions

**Comments to the Author**

1. If the authors have adequately addressed your comments raised in a previous round of review and you feel that this manuscript is now acceptable for publication, you may indicate that here to bypass the “Comments to the Author” section, enter your conflict of interest statement in the “Confidential to Editor” section, and submit your "Accept" recommendation.

Reviewer #1: All comments have been addressed

Reviewer #2: All comments have been addressed

2. Is the manuscript technically sound, and do the data support the conclusions?

Reviewer #1: (No Response)

Reviewer #2: Yes

3. Has the statistical analysis been performed appropriately and rigorously? 

Reviewer #1: (No Response)

Reviewer #2: Yes

4. Have the authors made all data underlying the findings in their manuscript fully available?

Reviewer #1: (No Response)

Reviewer #2: Yes

5. Is the manuscript presented in an intelligible fashion and written in standard English?

Reviewer #1: (No Response)

Reviewer #2: Yes

6. Review Comments to the Author

Reviewer #1: (No Response)

Reviewer #2: The reviewer greatly appreciates the changes that the authors have made to the originally submitted manuscript. This comment is just a suggestion. Table 1 could be merged with Table S1. As I said in my original comments I think Table S1 has very valuable information and that it should be in the body of the paper not in supplementary information.

7. PLOS authors have the option to publish the peer review history of their article (what does this mean?). If published, this will include your full peer review and any attached files.

Reviewer #1: No

Reviewer #2: No

---

## [Editor Report · Acceptance letter]

22 May 2020

PONE-D-20-08031R1 

One year cross-sectional study in adult and neonatal intensive care units reveals the bacterial and antimicrobial resistance genes profiles in patients and hospital surfaces 

Dear Dr. Oliveira:

I am pleased to inform you that your manuscript has been deemed suitable for publication in PLOS ONE. Congratulations! Your manuscript is now with our production department. 

With kind regards,

on behalf of

Dr. Zhi Ruan 

Academic Editor

PLOS ONE